# Amino Derivatives of Diaryl Pyrimidines and Azolopyrimidines as Protective Agents against LPS-Induced Acute Lung Injury

**DOI:** 10.3390/molecules28020741

**Published:** 2023-01-11

**Authors:** Alexander Spasov, Irina Ovchinnikova, Olga Fedorova, Yulia Titova, Denis Babkov, Vadim Kosolapov, Alexander Borisov, Elena Sokolova, Vladlen Klochkov, Maria Skripka, Yulia Velikorodnaya, Alexey Smirnov, Gennady Rusinov, Valery Charushin

**Affiliations:** 1Department of Pharmacology & Bioinformatics, Scientific Center for Innovative Drugs, Volgograd State Medical University, Volgograd 400131, Russia; 2I. Ya. Postovsky Institute of Organic Synthesis, Ural Branch of the Russian Academy of Sciences, Yekaterinburg 620108, Russia; 3Department of Technology & Organic Synthesis, Ural Federal University Named After the First President of Russia B. N. Yeltsin, Yekaterinburg 620002, Russia; 4Department of Organic and Biomolecular Chemistry, Ural Federal University Named after the First President of Russia B. N. Yeltsin, Yekaterinburg 620002, Russia

**Keywords:** pyrimidines, [1,2,4]triazolo[1,5-*a*]pyrimidines, inflammation, lung injury, cytokine release

## Abstract

The problem of lung damage originating from excessive inflammation and cytokine release during various types of infections remains relevant and stimulates the search for highly effective and safe drugs. The biological activity of the latter may be associated with the regulation of hyperactivation of certain immune cells and enzymes. Here, we propose the design and synthesis of amino derivatives of 4,6- and 5,7-diaryl substituted pyrimidines and [1,2,4]triazolo[1,5-*a*]pyrimidines as promising double-acting pharmacophores inhibiting IL-6 and NO. The anti-inflammatory activity of 14 target compounds was studied on isolated primary murine macrophages after LPS stimulation. Seven compounds were identified to inhibit the synthesis of nitric oxide and interleukin 6 at a concentration of 100 µM. The most active compounds are micromolar inhibitors of IL-6 secretion and NO synthesis, showing a minimal impact on innate immunity, unlike the reference drug dexamethasone, along with acceptable cytotoxicity. Evaluation in an animal model of acute lung injury proved the protective activity of compound **6e**, which was supported by biochemical, cytological and morphological markers.

## 1. Introduction

Acute lung injury in coronavirus [1] and other infections are associated with the hyperactivation of immune cells—in particular, macrophages. The nucleic acids of the virus activate the TLR2/8 receptors of the endosomes. TLR activation triggers intracellular signal transmission and induces the transcription of proinflammatory cytokines and inducible nitric oxide synthase (iNOS) genes via transcription factors NF-κB and AP-1 [2]. As a result, the release of cytokines (a cytokine storm) takes place. In particular, one of the key mediators of inflammation interleukin-6 (IL-6) and nitric oxide (NO) cause interstitial edema, oxidative damage and the death of lung cells. IL-6 is a cytokine involved in various inflammatory and immune reactions, cellular apoptosis and proliferation [3,4], and some oncogenic pathways [5]. Regarding iNOS, this isoform is not present in resting cells but is actively expressed during their activation in all types of tissues. After synthesis, iNOS is active for many hours or days and forms a large amount of nitric oxide. iNOS activity is associated with tissue damage in arthritis, nephritis, stroke, septic shock, etc. [2]. Thus, IL-6 and iNOS are important therapeutic targets. The regulation of their action is the basis of an effective strategy for the treatment of inflammatory conditions. Therefore, the development of new and selective inhibitors of IL-6 and NO overproduction is a relevant task for medicinal chemistry. Recent advances in the field of anti-inflammatory small molecules for the treatment of acute lung injury have been reviewed in [6].

Synthesis of low molecular weight biologically active compounds based on nitrogen-containing heterocyclic scaffolds is one of the promising approaches to design effective drugs. The desired selectivity can be achieved by substituent variations to realize chemical libraries according to combinatorial and molecular modeling methods. In the case of azines, heterocycles bearing aryl substituents that are in the *meta*-position relative to each other and carry hydrogen-binding polar groups are of particular interest, e.g., 3,5-diaryl substituted pyridines (**A**, Figure 1) with a V-shaped plane-conjugated geometry of the hydrocarbon skeleton have shown a noticeable inhibitory activity against interleukin-6 [7]. It has been shown that the presence of carboxy and hydroxy groups in the *para*-position of aryl rings leads to a higher activity. The antiproliferative activity of the second type of topoisomerase (topo IIa) has been studied for the series of 2,4-diphenyl-5,6-dihydrobenzo(*H*)quinoline-8-amines [8]. The compound with the hydroxy group in the *para*-position of the benzene ring (**B**, Figure 1) proved to exhibit a high inhibitory activity and specificity towards topo IIa. The fact that the formation of a plane-conjugated fragment of a molecule bearing maximally distant hydrophilic groups at the ends, apparently, provides the most effective interaction with the binding sites of the target, according to a 3D molecular modeling. A wide range of biological activity is demonstrated by 4,6-diaryl substituted pyrimidines [9] derived from chalcones—a biologically important class of natural compounds [10]. A high pharmacological potential of 5,7-substituted [1,2,4]triazolo[1,5-*a*]pyrimidines, including herbicidal, antifungal, antimalarial, antiviral, cardiovascular vasodilator, anti-inflammatory, analgesic, antimicrobial, and hypoglycemic activity, was demonstrated in the review [11]. Most of the structures considered comprise coordination aromatic and/or aliphatic hydroxy and amino groups. For example, an important coordinating role of NH groups was shown for 2-((1*H*-benzo[*d*]imidazol-2-yl)methylthio)-5-methyl-*N*-(4-(4-methylpiperazin-1-yl)phenyl)-[1,2,4]triazolo[1,5-*a*]pyrimidin-7-amine (**C**, Figure 1) in the binding and inhibition of histone–lysine-specific demethylase 1 (LSD1/KDM1A), involved in the development of cancer cells [12].

In addition, most of the low molecular weight inhibitors of nitric oxide synthase enzymes (iNOS, eNOS, and nNOS) are functionalized by various hydrogen-binding NH groups [13]. Moreover, the high selectivity, in particular, of aminopyridine inhibitors against iNOS compared to eNOS and nNOS, is associated with an increased length and conformational rigidity of their hydrocarbon skeleton (**D**, Figure 1). The mechanism of competitive binding to the active isoform site by molecules larger than *L*-arginine has been analyzed using the quantum chemical approach of anchor plasticity [14].

Thus, hydrophilic linker groups are important for modeling (fine tuning) the biological activity and selective action of these pharmacophores. The data regarding the biological activity of amino derivatives of 4,6- and 5,7-diaryl substituted pyrimidines and [1,2,4]triazolo[1,5-*a*]-pyrimidines are scarcely presented in the literature. Therefore, it appears to be of interest to apply the strategy of combining aniline substituents with a rigid heterocyclic scaffold to develop IL-6 and NO synthesis inhibitors. We report here the synthesis and pharmacological evaluation of anti-inflammatory activity of novel aminoaryl derivatives of pyrimidines and azolopyrimidines.

## 2. Results

### 2.1. Synthesis

To explore the structure-activity relationships in a series of 4,6- and 5,7-diaryl substituted pyrimidines and [1,2,4]triazolo[1,5-*a*]pyrimidines, aminoaryl-containing azines with different positions of amino groups in benzene rings and with various electron-donative substituents at C-2 position of the pyrimidine ring were synthesized. It is known that fluorine atoms may greatly impact biological activity [15], therefore several fluorine-containing azines have been synthesized. To determine the effect of the heteroaromatic core on the inhibitory activity of the studied pharmacophores, structurally similar 1,4-dihydropyrimidines and 4,7-dihydro-[1,2,4]triazolo[1,5-*a*]pyrimidines with aminoaryl fragments at the C-4 and C-7 positions of these heterocyclic systems, respectively, were obtained for comparison.

The three-stage synthesis of aminoaryl-containing azines from commercially available reagents is shown in Figure 2. At the first stage, the typical aldol condensation of aromatic aldehydes **1** with ketones **2** produced nitro-chalcones **3 [16]**. The synthesis of chalcones **3a**–**c** with one NO_2_ group under standard Claisen–Schmidt conditions provided moderate yields [17,18,19]. In the case of dinitro-substituted chalcones **3d**,**e**, the method using ultrasonic irradiation has been shown earlier [20] to give high yields of the reaction products. According to our research data, the condensation reaction, proceeding in the presence of H_3_BO_3_ (the Lewis acid) as the catalyst under reflux in acetic acid for 12 h, allowed us to obtain nitro-chalcones **3d**,**e** in 72–78% yields. These products were formed as pale yellow crystals at the end of the reaction, when the solutions were cooled. At the second stage, aromatic nitropyrimidines **4** or trialozopyrimidines **5** were obtained in 42–72% yields through the conjugated addition of substituted amidine or aminotriazole to chalcone in DMF, according to the previously described method [21]. Low yields of triazolopyrimidines **5** can be explained by the formation of by-products, such as 3-(3-amino-1*H*-1,2,4-triazol-1-yl)-1,3-diphenylprop-2-en-1-ones [22]. This is evidenced, in particular, by a singlet of the amino group at 5.98 ppm, a singlet of the proton of the prop-2-en-1-one fragment at 8.11 ppm and a significant upfield shift of the signals of the para-nitrobenzene fragment protons up to 8.17 and 6.71 ppm in the ^1^H NMR spectrum of 3-(3-amino-1*H*-1,2,4-triazol-1-yl)-3-(3-nitrophenyl)-1-(4-nitrophenyl)prop-2-en-1-one compared to product **7b** (Appendix A). Purification of **4** and **5** was carried out by chromatographic separation using chloroform as an eluent. Then, the nitro compounds were reduced by hydrazine hydrate in the presence of Raney nickel in ethanol or ethanol–THF mixture. The target amino derivatives of pyrimidines and triazolopyrimidines **6** and **7** were obtained (Appendix A).

The nitro-containing 1,4-dihydropyrimidines **8 [23]** and 4,7-dihydro[1,2,4]triazolo[1,5-*a*]-pyrimidines **9** [24,25] were obtained under conditions of the three-component Biginelli reaction (Figure 3). Further reduction of compounds **8**, **9** to amino derivatives **10** and **11** was carried out by action of hydrazine hydrate in the presence of Raney nickel in ethanol (Appendix A).

### 2.2. Target Compounds Inhibiting NO and IL-6 Release from LPS-Stimulated Macrophages In Vitro

Primary murine macrophages stimulated by *E. coli* LPS can be used as a phenotypic screening model for compounds preventing excessive inflammation. LPS in these cells activates TLR4 receptors, triggering intracellular signaling cascades via IRAK, BTK, Tyk, and JAK1/3 kinases that stimulate the synthesis of nitric oxide and proinflammatory cytokines. Initially, the activity of compounds was evaluated as the ability to inhibit the synthesis of NO, because it is the easiest and fastest way to assess the activation of macrophages. Compounds that inhibit the synthesis of NO by 40% or more at a concentration of 100 µM were considered to be promising ones. At the same time, they should not be cytotoxic for macrophages (cell viability >80% in the same concentration is desirable). Next, active compounds were studied in a wide range of concentrations to assess their influence on NO synthesis and IL-6 secretion [26,27]. The results of the screening are presented in Table 1. All synthesized heteroaromatic amino compounds **6** and **7** proved to exhibit a noticeable NO inhibitory activity, which virtually disappears on switching to structurally similar dihydropyrimidines **10** and triazolopyrimidines **11** (Table 1). These data appear to indicate that the flat-conjugated heteroaromatic system is one of the important factors that contribute to anti-inflammatory activity.

Analysis of the effect of the benzene ring substitution in azines **6** and **7** on inhibition of LPS-induced NO and IL-6 allowed us to draw the following conclusions. The presence of amino groups at C-2 and C-3 of aryl fragments of substituted azines **6a**, **d** and **7b** leads to a sharp decrease in inhibitory activity. On the contrary, the amino groups in the *para*-position of both aromatic rings in pyrimidine **6e** and [1,2,4]triazolo[1,5-*a*]pyrimidine **7c** provide the highest activity against NO and IL-6 secretion. The IC_50_ values for NO are comparable for **6e** and **7c** and have the values 16.24 and 15.20 µM, respectively. However, IL-6 inhibitory activity increases by an order of magnitude from **6e** to **7c** with the structural replacement of the pyrimidine ring by the azolopyrimidine core. The IC_50_ values for IL-6 for **6e** and **7c** proved to be 18.45 and 2.20 µM, respectively. Replacing one of the NH_2_ groups in the *para*-position of one of the benzene rings with a hydrogen atom in pyrimidine **6b** leads to a slight decrease in NO inhibitory activity and an increase in IL-6 inhibitory activity by an order of magnitude compared to **6e** (Table 1). On the contrary, replacing this amino group with a fluorine atom in pyrimidine **6c** and [1,2,4]triazolo[1,5-*a*]pyrimidine **7a** results in a significant two-order increase in NO inhibitory potency (IC_50_ 0.37 µM) and a decrease in IL-6 inhibitory activity compared to **6e** and **7c**, respectively. A sharp decrease in cytotoxicity observed in this case may indicate at the influence of the latter on NO synthesis more than on IL-6. In turn, the replacement of the methyl group in **6e** with the NH_2_ group in the pyrimidine ring of compound **6f** leads to a decrease in the overall anti-inflammatory activity. The introduction of the SCH_3_ group into the pyrimidine ring of compound **6g** provides a two-fold increase in NO inhibitory activity (IC_50_ 0.12 µM) and a complete loss of the activity against IL-6. Apparently, in this case, we deal with the selectivity of compound **6f** in respect to NO synthesis.

Thus, aniline substituted pyrimidines **6** and [1,2,4]triazolo[1,5-*a*]pyrimidines **7** can be considered as double-acting pharmacophores targeting IL-6 and NO. Compounds **6a**,**e** and **7a**,**c** demonstrate activity at the micromolar level. Factors, such as a plane-conjugated heteroaromatic scaffold, linker amino groups, and/or fluorine atoms in the *para*-position of benzene rings, of the methyl substituent in the pyrimidine ring appear to be decisive ones in increasing the efficiency and selective inhibitory action of the synthesized azines.

### 2.3. Active Compounds Demonstrate Variable Influence on Macrophage Phagocytosis

As the next step, we have examined how the most active compounds **6a**, **6e,** and **7c** affect the phagocytic activity of primary macrophages in comparison with the reference drug dexamethasone after 72 h of incubation (Figure 1). Using a microscopic examination, a number of active phagocytes and number of yeast cells in their cytoplasm per 100 macrophages were counted. Cell viability was determined as lactate dehydrogenase activity in cell culture media, which reflects impaired membrane permeability.

As shown in Figure 1, dexamethasone demonstrates a pronounced immunosuppressive activity consistent with the literature data, reducing both the number of phagocytes and their phagocytic capacity [28,29]. Compounds **6a** and **7c** preserve macrophage phagocytic activity but decrease the mean number of phagocytosed particles by 35% and 63%, respectively. In turn, compound **6e** decreases the phagocytic index by 32% but retains a nearly normal phagocytic capacity. No cytotoxic properties have been noted for the tested compound during the experiment. We may conclude that dexamethasone showed a pronounced immunosuppressive effect, while compound **6e** preserves the innate phagocytic activity of macrophages, being superior relative to compounds **6a** and **7c**.

### 2.4. Compound **6e** Protects LPS-Induced Acute Lung Injury In Vivo

Compound **6e**, which combines a marked anti-inflammatory activity in vitro with negligible suppression of phagocytosis and macrophage viability, has been tested in a murine model of LPS-induced acute lung injury in comparison with dexamethasone as the reference drug. Compounds **6a** and **7c** were more active as IL-6 inhibitors but also showed pronounced cytotoxicity in the MTT test and significantly suppressed macrophage phagocytic capacity; thus, they were excluded from further study. To assess the degree of inflammatory-related disorders, the behavioral, biochemical, cytological, and histological markers were assessed during 24 h after a single oropharyngeal administration of LPS.

#### 2.4.1. Open Field Test

Acute LPS-induced lung injury was accompanied with symptoms of neuroinflammation. We observed a decrease in motor, observational, and orientation activity of the LPS-treated animals in the “Open Field” test, which persisted for 24 h (Figure 2). These disorders were corrected by dexamethasone. Compound **6e** increased the motor activity of mice but was inferior to dexamethasone activity. The exploratory behavior was also improved, which reflects the antidepressant activity of the compound (Figure 3).

#### 2.4.2. Biochemical Markers of Inflammation

As shown in Figure 4, the administration of LPS resulted in a typical acute inflammation manifested in the secretion of IL-6 to the bronchoalveolar fluid and blood plasma. At the same time, the secretion of TNF-α was reduced at the end of the experiment consistently with the previous observations, which revealed that it peaks in 4–6 h after LPS administration [30,31]. The increased permeability of alveolar vessels for plasma proteins reflecting exudative inflammation was evident in the LPS control group (Figure 4). Both dexamethasone and compound **6e** normalized these markers of inflammation and lung damage. In the case of the lung permeability index, **6e** had the most significant effect, preserving the vascular permeability at intact levels. The level of TNF-α both in BAL and in blood plasma turned out to be an uninformative marker. Plasma IL-6 concentrations were highest in the LPS group, while dexamethasone and **6e** normalized it to intact levels. The BAL IL-6 concentration shows a minimal difference between experimental groups, but a protective trend for tested compounds is also evident.

#### 2.4.3. Leukocyte Markers

Examination of the blood and BAL leukocytes revealed that the local lung inflammation was more significant than the systemic one (Figure 5). The blood leukocyte population was comparable between groups. LPS treatment was associated with the recruitment of segmented neutrophils at the expense of lymphocytes. This was corrected by dexamethasone but not with compound **6e**.

The cellular composition of BAL also reflects an acute inflammatory process as a 3-fold increase in the content of mature segmented neutrophils, along with a corresponding drop in the content of monocytes (Figure 5). Mice treated with dexamethasone or **6e** demonstrated amelioration of this ratio, but it was only significant for the dexamethasone group.

In turn, dexamethasone proved the systemic immunosuppressive activity. It significantly decreased proliferation of spleen lymphocytes, while **6e** preserved normal content of spleen lymphocytes.

#### 2.4.4. Histological Markers

The development of LPS-induced inflammation was confirmed by histological examination. Purulent discharge pneumonia was evident, including pronounced infiltration of the interstitial lung tissue by polymorphonuclear neutrophils, the presence of purulent exudate in the lumen of the alveoli, thickening and swelling of the alveolar septa, and diapedesis of erythrocytes into the interalveolar septa. Due to the exudate clogging of alveolar lumen, dystelectases were detected in the undamaged parts of the lungs (Figure 6).

Treatment with **6e** limited inflammatory lesions, which were mainly characterized by intrauterine edema and infiltration of the interalveolar septa with leukocytes.

At the same time, it should be noted that, in all experimental animals of this group, despite the dense infiltration of immune cells into the lung tissue (Figure 7), alveolar septa damage was significantly less pronounced than in mice of the LPS control group, as shown in Table 2.

Thus, the synthesized compounds showed protective activity in vivo comparable to dexamethasone. An important feature of the studied compounds is that they do not cause immunosuppression, unlike dexamethasone and other steroid anti-inflammatory drugs.

## 3. Conclusions

There is an evident demand for development of novel effective and safe drugs against severe complications of infectious diseases, including COVID-19 and bacterial pneumonias. Previously, we described nitro-azolo[1,5-*a*]pyrimidines and quinazoline-2,4(1*H*,3*H*)-diones with anti-inflammatory and protective activity against LPS-induced acute lung injury [32,33]. The synthesis of amino derivatives of 4,6- and 5,7-diaryl substituted pyrimidines and [1,2,4]triazolo[1,5-*a*]-pyrimidines was carried out. The anti-inflammatory activity of 14 compounds was studied on primary peritoneal macrophages of C57BL/6J mice. Seven compounds were identified that significantly inhibit the synthesis of nitric oxide and interleukin 6 at a concentration of 100 µM. The most active compounds **6e**, **7a** and, especially, **6a**, **7c** were identified as micromolar inhibitors of IL-6 secretion and NO synthesis with acceptable cytotoxicity in the physiologically achievable concentration range. The influence of the nature and position of substituents in azine molecules on their inhibitory activity against IL-6 and NO has been analyzed. The following key factors affecting the effectiveness and selectivity of pharmacophores have been identified. These include the presence of a plane-conjugated heteroaromatic framework; linker amino groups; and/or fluorine atoms in the vapor position of benzene rings, a methyl substituent, or an azole fragment in the pyrimidine core. The most promising compound **6e** in an animal trial against LPS-induced pneumonia has shown activity comparable to that of dexamethasone. Unlike the latter, the designed compound does not cause immunosuppression in animals. Synthesized aniline substituted pyrimidines **6** and [1,2,4]triazolo[1,5-*a*]pyrimidines **7** can be considered as promising agents targeting excessive IL-6 and NO secretion under inflammatory conditions.

## 4. Materials and Methods

### 4.1. Synthesis

Commercial reagents were obtained from Sigma-Aldrich (St. Louis, MO, USA), Acros Organics (Geel, Belgium), or Alfa Aesar (Ward Hill, MA, USA) and used without any further purification. All workup and purification procedures were carried out using analytical-grade solvents. The synthesis of compounds **3**–**11** and their physical characteristics are given in Appendix A.

Test compounds were dissolved in 99% DMSO (stock concentration 40 mM) and stored at −25 °C. If sediment or opalescence was detected, 5% *v*/*v* Tween 20 (Merck) was added. Serial dilutions were prepared ex tempore in a media suitable for the particular study. The final concentration in samples: DMSO < 0.25%, Tween 20 < 0.025% were added to control samples in equal concentrations.

### 4.2. Animals

All procedures with animals in the study were carried out under the generally accepted ethical standards for the manipulations of animals adopted by the European Convention for the Protection of Vertebrate Animals used for Experimental and Other Scientific Purposes (1986) and taking into account the International Recommendations of the European Convention for the Protection of Vertebrate Animals used for Experimental research (1997). All sections of this study adhere to the ARRIVE Guidelines for reporting animal research [34]. Male mice (21–24 g) were housed 5 per cage in ambient lighting and 60% humidity. Animals had free access to water and food before the study.

### 4.3. Isolation and Treatment of Peritoneal Macrophages

Peritoneal macrophages (PM) were isolated from the peritoneal exudate of 30 male C57bl/6j mice. To accumulate PM, 1 mL of 3% peptone solution was injected intraperitoneally. After 3 days, the mice were euthanized by cervical dislocation. Cells of peritoneal exudate were obtained by aseptic washing of the abdominal cavity with 5 mL of sterile Hanks’s solution (+4–6 °C) without calcium and magnesium ions. The total number and viability of cells were assessed in a Goryaev counting chamber (Russia) with a 0.4% trypan blue staining (Sigma-Aldrich, St. Louis, MO, USA). The cell concentration was adjusted to 1.0 × 10^6^ cells/mL in DMEM (Gibco, Waltham, MA, USA) supplemented with 2 mM l-glutamine (Gibco), 10% heat-inactivated fetal bovine serum (BioClot, Bavaria, Germany), 100 U/mL penicillin, and 100 mg/mL streptomycin (Gibco) and plated 200 µL/well in 96-well transparent plates (SPL Life Sciences Co., Ltd., Pocheon-si, Republic of Korea). After 2 h at 37 °C in a humidified atmosphere with 5% CO_2_, the wells were washed to remove non-adherent cells. After 24 h of incubation, 20 µL of the supernatants were substituted with 20 µL of solutions of test compounds, followed by *E. coli* O127:B8 LPS (100 ng/mL final concentration) after 30 min. The experiments were run in 3 independent replicates.

### 4.4. Assay of Nitric Oxide (NO)

The accumulation of the nitrite anion (a stable end product of NO decomposition) in supernatants was determined using a standard Griess reagent. Briefly, 50 µL of supernatants were collected 22 h after incubation of PM, and the test and control compounds were mixed with 50 µL of 1% sulfonamide in 2.5% H_3_PO_4_ and 50 µL of 0.1% *N*-(1-naphthyl) ethylenediamine in 2.5% H_3_PO_4_. After incubation at 23 °C for 10 min in an orbital shaker, the optical density was determined at a wavelength of 550 nm with a microplate reader Infinite M200 PRO (Tecan, Grödig, Austria).

### 4.5. Assay of Cytokines

The cell supernatant was collected and centrifuged at 1000× *g* for 20 min in a 2–16 PK centrifuge (Sigma, Osterode am Harz, Germany). The concentrations of IL-6 and TNF-alpha were determined by ELISA using commercial kits (Cloud-clone, Houston, TX, USA) with a microplate reader Infinite M200 PRO (Tecan, Grödig, Austria).

### 4.6. Cytotoxicity Study

The activity of lactate dehydrogenase (LDH) in a cell culture medium served as a marker of membrane permeability and cell death. Aliquots of supernatants were taken after 24 h of inoculation with test compounds, mixed with 250 µL of 0.194 nM NADH solution in 54 mM phosphate-buffered saline (pH 7.5). Then, 25 µL of a 6.48 mM pyruvate solution was added to the mixture. The optical density was followed at a wavelength of 340 nm for 20 min. The MTT test was performed 24 h after incubations of cells with tested compounds. Briefly, 20 µL of MTT solution was added to each well and incubated at 37 °C in a humidified atmosphere containing 5% CO_2_ for 4 h. The culture medium was removed, the cells were lysed, and formazan crystals were dissolved in 150 µL DMSO. The plates were shaken at room temperature for 10 min, and the optical density was measured in a microplate reader Infinite M200 PRO (Tecan, Grödig, Austria) at a wavelength of 565 nm.

### 4.7. Phagocytosis Assay

Peritoneal macrophages of C57bl/6j mice were cultured in a 24-well plate at a volume of 500 µL/well (1 × 10^5^ cells/mL) for 24 h. Test compounds or DMSO were added to wells in triplicate followed by 50 µL of 1% yeast suspension, and the plates were incubated for 40 min. After 24 or 72 h, the medium was removed. The cells were fixed with May–Grunwald methanol dye and stained with Azur–Eosin in Romanovsky’s modification for 45 min at room temperature, washed, air-dried, and analyzed using a microscope Mikmed-6 (Russia), equipped with a digital camera. The study was performed in three technical replicates and two series; 100 macrophages were processed in each sample to count the captured yeast cells. Macrophage spreading was assessed as a number of pseudopodia.

### 4.8. LPS-Induced Acute Lung Injury

The randomization of C57BL/6J mice was performed by body weight and motor activity in an open field test. Dexamethasone (5 mg/kg) and **6e** (30 mg/kg) were administered intraperitoneally in 10 mL/kg sterile saline. The control animals received an equal volume of the vehicle. After 1 h, the mice were anesthetized with isoflurane inhalation until their breathing rate was decreased. The mice were suspended by the front incisors on an inclined surgical table, the tongue was pulled out with narrow curved tweezers, and 1 mg/mL of *E. coli* O127:B8 LPS (Sigma-Aldrich, St. Louis, MO, USA) in 1 mL/kg sterile saline was instilled into the back of the oropharynx [35]. Intact control animals received an equal volume of sterile saline similarly.

### 4.9. Open Field Test

The animals were placed in the center of a round open field with an arena of 44 cm diameter and 32 cm wall height under 300 lux lighting. During 5 min of observation, horizontal motor activity, vertical motor activity, time spent in the central part, the number of exits to the central part, and exploratory activity parameters were recorded. Body temperature was determined rectally.

### 4.10. Bronchoalveolar Lavage and Plasma Preparation

The mice were anesthetized with 500 mg/kg chloral hydrate (Sigma-Aldrich, St. Louis, MO, USA) intraperitoneally 24 h after LPS administration. Blood was taken by intracardiac puncture in test tubes with heparin. The blood samples were centrifuged at 1000× *g* and 4 °C for 15 min on a 2–16 PK centrifuge (Sigma, Germany), and plasma was separated and stored at −80 °C until the assay. Thoracotomy was performed, the ligature was applied to the left bronchus and the trachea was cannulated with a 20 G needle. The right lung was washed twice with 0.7 mL of warm sterile saline. After combining aliquots, the bronchoalveolar lavage (BAL) was centrifuged at 800× *g* and 4 °C for 10 min. The supernatant was separated and stored at −80 °C until the assay. The residual cell pellet was resuspended in 50 µL of PBS for further study. The left lung was washed in saline and placed in 10% buffered formalin for morphological evaluation.

### 4.11. Leukocyte Count in Blood and BAL

The total number of leukocytes in heparinized blood was determined after staining with methylene blue in Goryaev’s counting chamber at ×100 magnification. Smears of blood and BAL cell pellets were air-dried, fixed according to May–Grunwald for 3 min, and stained according to Romanowsky–Giemsa. After 30 min, the slides were washed, air-dried, and examined with an immersion objective (×1000 total magnification). On each slide, a total of at least 100 cells were calculated.

### 4.12. Lung Permeability Index

The concentration of total protein in BAL supernatants was determined spectrophotometrically with the pyrogallol red method, and the protein in the blood plasma was determined with the biuret method using commercial kits (Vital, Saint Petersburg, Russia) and bovine serum albumin as the standard. The lung permeability index was calculated as the protein concentration in the BAL to plasma ratio.

### 4.13. Histological Study

Morphological markers of inflammation in lung tissue were assessed in a semi-quantitative way [36] on paraffin sections stained with hematoxylin and eosin. The sections were examined under a light microscope (Zeiss, Germany) by a double-blind method. The score of inflammation was determined as follows: 0—the presence of single inflammatory cells; 1—weak inflammation, inflammatory cells infiltrate no more than 10% of the lung tissue, including interalveolar cell partitions; 2—moderate inflammation, inflammatory cells infiltrate no more than 50% of the structures of the lung, but the interstitial tissue is identified; and 3—severe inflammation, inflammatory cells densely infiltrate more than 50% of the lung tissue and airways. At the same time, lymphoid follicles localized around large and medium bronchi were not taken into account in the final assessment. Paraffin sections of 5 um thickness were mounted on slides treated with poly-L-lysine (Menzel GmbH & Co. KG, Braunschweig, Germany). After dewaxing and rehydration, they were incubated in 3% hydrogen peroxide for 20 min to block endogenous peroxidase. Immunostaining was carried out using the MAX PRO (MULTI) peroxidase–polymer imaging system according to the manufacturer’s instructions (Histofine). The unmasking of antibodies was carried out by boiling sections at 100 °C in 0.01 M citrate buffer (pH 6.0) for 20 min. Sections of lung tissue were incubated with primary antibodies to CD68 (clone 3F-103, Santa Cruz Biotechnology, Dallas, TX, USA) at room temperature for 1 h and treated with 3,3′-diaminobenzidine. Finally, the sections were stained with Mayer’s hematoxylin. The slides were studied and photographed using an AxioScope.A1 microscope (Zeiss, Munich, Germany) equipped with an AxioCam MRc5 camera. The photos obtained were processed using ZENpro 2012 (Zeiss).

### 4.14. Data Analysis

Statistical analysis and graph preparation were performed in Prism 7.0 (GraphPad Software Inc., San Diego, CA, USA). 1-Way ANOVA with a Dunnett’s post-test was used for multiple comparisons and Mann–Whitney *U* test for pairwise comparisons. IC_50_ values were calculated with nonlinear 3-parametric regression.

## Data Availability

Data are contained within the article.

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
