# Peer review of "Amino Derivatives of Diaryl Pyrimidines and Azolopyrimidines as Protective Agents against LPS-Induced Acute Lung Injury"

_molecules, 2023, doi:10.3390/molecules28020741_

Round 1

Reviewer 1 Report

The activity of the activity compounds seem not very significant, more importantly, LPS-induced ALI model is a classical model for the acute inflammatory, why author didn't see any change even in model group( es TNFa);

Authore should proveide a figure to demonstrate the design of their compounds.

lead compound usually  means the compound which you modify, the compound author obtained should call it active compound or something.

RAW image of NMR should be provided.

In figure 5 legend, what is # represented?

Author Response

Reviewer 1

Response

The activity of the activity compounds seem not very significant, more importantly, LPS-induced ALI model is a classical model for the acute inflammatory, why author didn't see any change even in model group( es TNFa);

Protective activity of compounds with similar micromolar levels of activity is known from the literature, e.g. Wang, B. S., Huang, X., Chen, L. Z., Liu, M. M., & Shi, J. B. (2019). Design and synthesis of novel pyrazolo [4, 3-d] pyrimidines as potential therapeutic agents for acute lung injury. Journal of enzyme inhibition and medicinal chemistry, 34(1), 1121-1130. In our study development of LPS-induced ALI was confirmed by marked increase in vascular permeability index, IL-6 levels, most importantly, by massive infiltration of the lung tissue with immune cells and histologically confirmed lesions. Low concentrations of TNF-alpha 24 hours after LPS treatment is consistent with data from Faffe, D. S., Seidl, V. R., Chagas, P. S., de Moraes, V. G., Capelozzi, V. L., Rocco, P. R., & Zin, W. A. (2000). Respiratory effects of lipopolysaccharide-induced inflammatory lung injury in mice. European Respiratory Journal, 15(1), 85-91 and Simons, R. K., Junger, W. G., Loomis, W. H., & Hoyt, D. B. (1996). Acute lung injury in endotoxemic rats is associated with sustained circulating IL-6 levels and intrapulmonary CINC activity and neutrophil recruitment—role of circulating TNF-± AND IL-± 6. Shock, 6(1), 39-45.

Authore should proveide a figure to demonstrate the design of their compounds.

The Scheme 1 showing the design of our compounds is provided.

lead compound usually means the compound which you modify, the compound author obtained should call it active compound or something.

Thank you for pointing this out, we replaced «lead compound» with «the most active/promosing» throughout the manuscript.

RAW image of NMR should be provided.

Synthesis schemes and 1H NMR spectra of the target anilines have been included in the Supplementary Materials.

In figure 5 legend, what is # represented?

It is a statistical significance vs. LPS group. Figure 5 caption was supplemented accordingly.

Reviewer 2 Report

The authors have designed and synthesized a series of 4,6- and 5,7-diaryl substitute pyrimidines and [1,2,4]triazolo[1,5-a]pyrimidine, which were subjected to evaluate the anti-inflammatory activity on isolated primary murine macrophages after LPS-stimulation. Several compounds are micromolar inhibitors of IL-6 secretion and NO synthesis, showing a minimal impact on innate immunity, along with acceptable cytotoxicity. The lead compound 6e has shown the protective activity in animal trial against LPS-induced acute lung injury, which was supported by biochemical, cytological and morphological marker. I will recommend publication on Molecules after following minor revisions.

1. The structures of aromatic aldehydes 1 and compouds 8-11 are incorrect. 

2. Pls provide the NMR spectrum of the target compounds in SI.

3.  The introduction of the compounds or drugs for treating the acute lung Injury (ALI) is insufficient in this manuscript. The following review of the small molecules with anti-inflammatory activities for the treatment of ALI is suggested to be cited in the introduction. (Eur. J. Med. Chem., 2020, 207,112660

Author Response

Reviewer 2

Response

The structures of aromatic aldehydes 1 and compouds 8-11 are incorrect.

The structures of aromatic aldehydes 1 have been corrected in Schemes 2.1 and 2.2. The structures of compounds 8-11 have been corrected in Scheme 2.2 and also in theTable 2.1

Pls provide the NMR spectrum of the target compounds in SI.

1H NMR spectra of the target anilines have been added in the Supplementary Materials. We have eliminated the inconsistency in the numbering of compounds 4c, 4d, 4e in the Supplementary Materials by swapping the description of these compounds, and also corrected some typos.

The introduction of the compounds or drugs for treating the acute lung Injury (ALI) is insufficient in this manuscript. The following review of the small molecules with anti-inflammatory activities for the treatment of ALI is suggested to be cited in the introduction. (Eur. J. Med. Chem., 2020, 207,112660)

Thank you a lot, we modified the introduction section accordingly.

Round 2

Reviewer 2 Report

The authors have revised the manuscript. This research will be recommended to be published after the following minor revision.

1.  This work is related to  the literature reported by Chen, T. P. et. al (Eur. J. Med. Chem.2021, 225, 113766), which described a series of the 2-arylamimo pyrimidines against the LPS-induced acute lung injury.  Therefore, this literature is suggested to  be cited in the appropriate position. 

2.  In SI, the m.p. (276-127 °C) of compound 11b is wrong. 

3.  The 1HNMR spectrums attached are not recorded adequately. The chemical shift value  and integrated value are missed.  Additionally, Pls add the 13CNMR spectrum of the corresponding compounds in the SI. There is some wrong with 1HNMR analysis of compound 11a and 11b. Pls check carefully the 1HNMR and 13CNMR spectra data of all the compounds.